# An Allelopathic Role for Garlic Root Exudates in the Regulation of Carbohydrate Metabolism in Cucumber in a Hydroponic Co-Culture System

**DOI:** 10.3390/plants9010045

**Published:** 2019-12-27

**Authors:** Haiyan Ding, Ahmad Ali, Zhihui Cheng

**Affiliations:** 1School of public health, Dali University, Dali 671000, Yunnan, China; woaimama195710@nwsuaf.edu.cn; 2College of Horticulture, Northwest A&F University, Yangling 712100, Shaanxi, China; ahmadhort87@nwsuaf.edu.cn

**Keywords:** allelochemicals, garlic root exudates, oxidative stress, chlorophyll fluorescence, carbohydrate metabolism

## Abstract

Garlic is considered to have a strong positive effect on the growth and yield of receptors under soil cultivation conditions. However, how this positive promotion is produced by changing the growth environment of the receptors or directly acting on the receptors is still not very clear. The direct influence of co-culturing with different quantities of garlic plants (the control 5, 10, 15, 20) on the growth and biochemical processes of cucumber plants was studied using a hydroponic co-culture system. Different numbers of garlic bulbs inhibited the growth of cucumber plants and increased the production and induction of reactive oxygen species, which accompanied the enhancement of lipid peroxidation and oxidative damage to cucumber. This allelopathic exposure further reduced the chlorophyll contents and photosynthesis rate, and consequently impaired the photosynthetic performance of photosystem II (PSII). Garlic root exudates increased the leaves’ carbohydrates accumulation, such as soluble sugar contents and sucrose levels by regulating the activities of metabolismic enzymes; however, no such accumulation was observed in the roots. Our results suggested that garlic root exudates can mediate negative plant–plant interactions and its phytotoxic influence on cucumber plants may have occurred through the application of oxidative stress, which consequently imbalanced the source-to-sink photo-assimilate flow.

## 1. Introduction

Chemical-mediated plant–plant interference is an important allelopathic mechanism by which certain plant species produce and release organic compounds into the growth environment through leaching, volatilization, root exudation, and passive discharge through decomposition, which have beneficial and harmful effects from the donor plant on the receptor plant [1]. Many plant organs, such as root tissues or leachates, play diverse physiological functions in term of resource competition and root-mediated rhizospheric interactions that have strong allelopathic significance by involving the secretion of potent phytotoxins or organic chemicals, defined as allelochemicals [2]. Plant root exudates represent the dynamic source of allochemicals secretion, so it can mediate plant–plant interactions only at sufficient concentrations to affect plant growth and development [3,4]. The concentration of allelochemicals varies with the biological and non-biological cues; therefore, their functions and goals are different. A series of action modes have been reported as being allelochemical, such as those influencing the cell division and elongation [1], destroying the integrity and function of the cell membrane [5], affecting water status and nutrient uptake of the plant [6], altering photosynthetic and respiratory chain electron transport [7], interfering with the plant hormone balance [8], and altering the synthesis and activity of plant proteins and gene expression [9], thereby affecting the growth and development of plants.

The negative response of receptors to allelochemicals manifests as growth retardation and tissue destruction, which results in oxidative damage to the plant system, where oxidative stress is thought to be a putative mode of action of allelochemicals induced by the overproduction of reactive oxygen species, and consequently causes lipid peroxidation [10]. The most frequently observed physiological effects associated with ROS-mediated damage triggered by alleopathic compounds are disturbances of photosynthesis and the diminished amplitude of chlorophyll a variable fluorescence [11]. The major coordinate changes induced by oxidative stress are the down-regulation of photosynthetic machinery, impairment of the carbohydrate metabolism, and interruption of the source-to-sink transport flow of assimilates [12]. The inhibition of photo-assimilate output to the sink tissues results in the accumulation of soluble sugars and sucrose in source leaves, which leads to the inhibition of photosynthetic feedback [13].

Cucumber is an important economic vegetable mostly grown under greenhouse cultivation in China. However, consecutive continuous cropping obstacles and allelopathic autotoxicity of this crop have become severe threats to cucumber production, which is accredited to the accumulation of phytotoxic phenolic compounds in the substrates used for long-time cultivation [14,15]. Garlic is considered to be a strong allelopathic plant in field multi-cropping systems in China, where it is often used to multi-crop with other economic crops like cucumber and tomato to alleviate their continuous cropping obstacles by farmers in field production [16,17]. What is more, many researchers claim that garlic has a positive effect on the growth and yield of receptors under soil cultivation conditions by effectively changing their growth environment, such as improving the biochemical characteristics of the culture medium, thereby reducing the obstacles for continuous vegetable production [16,18]. 

However, some bioassay experiments advocated that garlic produced a direct negative and dose-dependent effect on receptors under controlled laboratory environmental conditions [19,20]. These studies indicate that it remains unclear whether this is a direct action of garlic root exudate or simply its effect on environmental regulation. The aim of this study was to investigate the direct mechanism of garlic root exudates on the growth and biochemical processes of cucumber under controlled laboratory conditions. In this study, a hydroponic co-culture system was adopted to confirm the effects induced by garlic root exudates on the growth, oxidative damage, and chlorophyll fluorescence and carbohydrate metabolism in cucumber plants, which can exclude possible nutritional and microbial disturbances when assessing the allelopathic potential of garlic. This study may provide important implications for understanding the mechanism of garlic root exudates on receptors and provide a new understanding of how garlic operates in field production.

## 2. Results

### 2.1. Plant Growth Parameters 

The present study revealed that the growth of cucumber plants was significantly suppressed with an increase in the cucumber/garlic (C/G) ratios when compared to the control, as shown in Table 1. Both the root and shoot dry mass accumulation and leaf area expansion of the cucumber seedlings were significantly inhibited with increasing intensities of garlic root exudates (*p* < 0.05), and the magnitude of this inhibition was also markedly apparent for the total dry mass of the plant. The SPAD (soil plant analysis development) value of each treatment in all measurement periods was lower than that of the control (Figure 1). However, there was no significant difference among the four treatments after co-culturing for 36 days.

### 2.2. Oxidative Damage Parameters

Garlic root exudates exerted a differential oxidative stress in cucumber leaves and the intensity of oxidative damage caused by malondialdehyde (MDA) and proline accumulation was remarkable, showing a clear dose–response relationship (Figure 2). After co-culturing for 36 days, the content of MDA and proline in the cucumber leaves increased with the increase of treatment concentration and reached the maximum at a C/G ratio of 2:20, which was an increase of 120.3% (MDA content) and 68.7% (proline content) compared to the control. However, the degree of damage imposed by treatments at an early stage was not severe but become virulent with exposure time compared to that in the control.

Another deleterious effect of garlic root exudates exposed with different number of garlic bulbs in the growth medium was the production of reactive oxygen species (ROS). The data showed that the effects on H_2_O_2_ content in cucumber tissues were more obvious and had relatively significant effects on superoxide anions (Figure 3). Dramatically, superoxide anion contents did not increase in proportion to the increases in H_2_O_2_ content for the 28 and 36 days of evaluation. However, the levels of superoxide anions in both the control and treated tissue were consistently similar, even when co-cultured for 28 and 36 days (Figure 3B). 

### 2.3. Protective Enzyme Activities

The activity of three kinds of leaf cucumber cell protective enzymes showed an upward trend at each stage of assessment (Figure 4). However, a dose-dependent effect was not greater in magnitude but was comparatively significant relative to the control in most cases. For comparative evaluation among treatments, the activity of SOD (superoxide dismutase, EC 1.15.1.1) and CAT (catalase, EC 1.11.1.6) at 20 days, and POD (peroxidase, EC 1.11.1.7) at 28 and 36 days, was found to be relatively similar after garlic induction, but each treatment progressively increased for all the enzymatic activities during the cucumber and garlic co-cultured hydroponic experiment. 

### 2.4. Non-Enzymatic Antioxidant Compound Content 

The experiment revealed that the investigated cucumber seedlings exposed with 2:0, 2:5, 2:10, 2:15, and 2:20 ratios had a significant observed difference in ascorbate (ASA) and glutathione (GSH) synthesis compared with the control (Figure 5). However, both ASA and GSH formation was not synthesized with a great tendency during the early stages, but finally at 36 days of measurement, the activity of all garlic treatments showed a differential response to some extent.

### 2.5. Gas Exchange Parameters

The effect of garlic root exudates on photosynthetic gas exchange parameters is shown in Figure 6. The examined treatments did not promote photosynthetic activity, but instead resulted in an inhibited net photosynthetic rate for cucumber leaves with a differential exposure to 20 days. Expectedly, the reduction rate due to the garlic concentration was altered with time, where the severity of inhibition in the net photosynthetic rater (Pn), stomatal conductance (Gs), and transpiration rate (Tr) in the early vegetative stage was significant but low, which changed to become more complex and severe with no marked difference at the reproductive stages. Moreover, no significant difference was observed between the 2:0 and 2:20 C/G treatment ratios for 28 and 36 days of exposure. The net CO_2_ assimilation rate changed with plant age in cucumber leaves but was not statistically significant throughout the whole experiment.

The chlorophyll fluorescence analysis indicated that the photosynthetic performance of the cucumber plants was dramatically impaired by the garlic bulb infusion (Table 2). The significant substantial damage of photosystem II (PSII) emerged at 20 days after treatment but the reduction rate was not evident during next evaluation times on the 28th and 36th days after the co-culture. Consequently, the Fv/Fm (photochemical efficiency of PSII), ΦPSII (quantum yield of PSII photochemistry), Fv’/Fm’(capture efficiency of PSII) and qP(photochemical quenching coefficient) values of both the control and treatment appeared with same difference. On the contrary, the non-photochemical quenching (NPQ) value of each treatment of cucumber leaf was significantly higher (*p* < 0.05) than that of the control and the extent of NPQ progress increased with increased concentration only after 20 days of exposure. 

### 2.6. Soluble Sugar and Starch Contents

In order to investigate the garlic root exudate’s effect on the cucumber carbohydrate metabolism, we determined the total soluble sugar, hexose, starch, and sucrose in the root and leaf samples (Figure 7). Variations in both the roots and leaves of cucumber seedling showed there was an antipodal carbohydrate regulation pattern. In the root samples, a dose-dependent effect did not induce any significant change at any tested duration, but the concentration ofhexose and starch was improved in all treatments on the 20th day of co-culture. Surprisingly, the amount of these carbohydrates was found to be higher in the control than in the treatments after 28 and 36 days of exposure. No differences were found between the control and garlic-treated plants concerning total soluble sugar (TSS) and sucrose concentrations after 20 days of co-culture. On the other hand, in cucumber leaves, the accumulation of total soluble sugar, sucrose, and hexose were observed to be higher than the control at each growing stage. Noticeably, the level of these carbohydrates changed due to treatments with a slight difference between them, especially after 36 days of evaluation.

### 2.7. Activities of Sucrose-Metabolized Enzymes in Leaves

Garlic root exudates with different concentrations reduced the plant acid invertase (AI) and sucrose synthetase–cleavage (SS-c) activities (Figure 8). The drop in amplitude of both enzyme contents with the number of garlic bulbs was not large but was significantly greater than that of normal cucumber seedlings for different durations. In contrast, the sucrose synthetase–synthetic (SS-s) and sucrose phosphate synthase (SPS) activities were significantly higher compared with the control throughout the co-culture period. However, the higher concentrations imposed a marginal effect and their observed influence was relatively constant relative to each other when the cucumber plants were evaluated at various points.

## 3. Discussion

Root exudate is one of the important ways to release plant allelochemicals; however, the complexity of its action environment has been difficult to study and understand. Different substances in the root exudate play different functions and have different effects; therefore, the extraction of a single substance for research has certain limitations [21,22]. For this reason, a hydroponic co-culture system was used to simulate the inter-cropping mode of garlic and other economic crops. Different numbers of garlic plants were co-cultured with cucumber plants under strictly controlled conditions to further clarify the influence of different concentrations of garlic root exudates on the physiological activities of cucumber during its vegetative growth stage.

### 3.1. Effect on Morphological Traits

Our results indicate that garlic species display a strong allelopathic activity, which significantly inhibits the growth of cucumber plants by imposing oxidative stress; its root exudates can release some growth inhibitors or allelochemicals, which can reduce the growth of target plants. This further confirmed our previous hypothesis that garlic root exudates not only impaired the morphological indices of pepper plant, but also altered the plant physiology by indirectly increasing the incidence of oxidative damage [19]. Aqueous garlic extract (AGE) contains allicin as the main antimicrobial substance, and its dose-dependent response can be explained by the synergy and antagonism of AGE on the growth of the cucumber plants [20]. Diallyl disulfide (DADS) is another potent allelochemical derived from garlic root exudates, where its exogenous application has performed a range of phytotoxic activity for tomato root growth inhibition, possibly due to cell division, changes in phytohormone contents, and changes in the expression levels of expansion genes [23]. Moreover, phytotoxic investigations have also shown the dose-dependent mode of action, where these findings also advocated an increase in intensity with increasing concentrations of extracts or root exudates, and a remarkable inhibitory effect was observed at the highest concentration [24,25]. 

In this study, the significant growth difference among different treatments could be attributed to the garlic root exudates involved in the endogenous hormonal imbalance by producing allelochemical stress, which strongly affected the cucumber growth due to the growth environment and nutrient concentration during the whole growth period for cucumber plants that were apparently identical [26]. However, the no difference in synergistic effect was found in the mode of action for cucumber growth inhibition, which was in a dose-dependent but antagonistic manner, when compared with our earlier garlic–pepper hydroponic co-cultured study. A low concentration of garlic root exudates had a growth promotion effect, but high concentrations exhibited a deleterious effect on pepper plants [19]. In contrast, all quantities of garlic bulbs exhibited a phytotoxic effect on cucumber growth without any treatment distinction compared to the control. 

### 3.2. Effect on the Antioxidant System

After the phenotypic malformation of plants begins, a further imbalance of plant biochemical and physiological entities may maturate through the imposition of a wide range of oxidative damage and osmotic stress. Stressed plants respond according to biotic and abiotic fluctuations; however, variable resistance or adaptation mechanisms depend on the stress tolerance degree, growth, exposure duration, and different developmental stages, which increase their complexity [2]. The oxidative stress based on allelopathy reflects the imbalance of the metabolic system caused by the excessive production and accumulation of reactive oxygen species, lipid membrane peroxidation exacerbates and cell membrane structure damage. The production of ROS and the induction of oxidative stress are among the putative modes of action of many allelochemicals [27]. 

In this study, we measured the oxidative stress by means of lipid peroxidation and proline synthesis, and the results showed the cucumber’s sensitivity to oxidative and osmotic damage. The MDA appeared as a primary stress indicator that reflects the amount of membrane damage in cucumber caused by garlic root exudates, and the degree of damage was dose-dependent. There were quantitative and co-culture time differences, and the amplitude of the membrane lipid peroxidation increased linearly with the increasing quantity of garlic bulbs (Figure 2A). Numerous findings suggested that the MDA level increased in response to allelochemical stress, akin to the oxidative stress in tomato seedling [28] and fig leaf gourd [14], through free radical formation in the plasma membrane. Enhanced MDA further suggested that the allelopathic root exudate induced oxidative stress, leading to disruption of the cell membrane structure, loss of cell integrity, and ROS formation [29]. Consistent with our findings, evidence shows that the root exudate or aqueous extract of garlic applied in higher concentrations significantly increased the MDA contents in the receiver vegetables crops [19,30]. Proline is responsible for osmotic regulation of plants under stress conditions and serves as an osmoprotectant. It can also act as an ROS scavenger as the main component of the antioxidative network and participate in reducing stress-induced oxidation [31,32]. Despite their well-known role in osmoregulation, the dual function of proline catabolism is also measured in triggering signal molecules such as ROS; rapid proline accumulation involved in the initiation of programed cell death is also a sign of oxidative damage [33]. Interestingly, in our study, the increment of MDA contents in treated garlic root exudates was proportional to the increase of proline production, validating the dual function of proline. One possibility is that the lipid peroxidation levels may indicate osmotic stress due to an increased proline content, and the adoptive response of proline could play a stress tolerance role [34]. Another hypothetical assumption about oxidative damage by proline via ROS production is more cognate in this situation. To follow this speculation, our results also indicate that despite an increase in proline accumulation with increasing phenolic concentration, proline accumulation was a consequence of cellular injury and protein damage [35]. Exposure to major components of volatiles, also indicate that increased levels of H_2_O_2_, MDA, and proline are signs of lipid peroxidation and the induction of oxidative stress [36,37]. The incorporation of sugarcane straw leachate induced proline accumulation in the roots and cotyledons of arrow leaf. The proline increase was related to oxidative stress in the roots but not in the cotyledons [38].

Intracellular origins of reactive oxygen species (ROS) and modification of the redox state or redox homeostasis in the stressful plant community was identified as a common allelopathic reaction [39]. In many cases, the impact of allelochemicals on ROS-related oxidative imbalance resulted in cell wall alteration, protein oxidation, DNA or RNA cleavage, photosystem inhibition, and the inactivity of ROS scavengers [40]. In the present study, garlic root exudates exhibited the strong phytotoxic occurrence in an over-accumulation of ROS (O^2−^ and H_2_O_2_) in cucumber leaves, which increased the membrane lipid peroxidation and caused cellular damage. Consistent with these results, the plant response after exposure to allelochemicals was equivalent to that for allelopathic stress, which increased the accumulation of oxygen derivatives (ROS), such as the superoxide radical O^2–^ anion and H_2_O_2_ in plant cells [41]. The increased H_2_O_2_ and OH^−^ production might be correlated with the lipid peroxidation observed in cucumber leaves [34].

When the ROS-mediated oxidative stress becomes severe and prolonged, plant cells respond against the adversarial ecosystem. The defensive metabolic systems are armed with the antioxidant machinery consisting of enzymatic components and non-enzymatic antioxidants, which play an important role in ROS detoxification. ROS detoxification and increases of ROS-scavenging enzyme activities accompanied by allelochemicals has been observed in cucumber roots [10]. Figure 4 shows that all these enzyme activities significantly increased with various ratios of garlic root exudates compared to the control. In comparison, the CAT activity changed with significant co-culturing time differences, and a dose-dependent difference was not seen at the early stage but gradually increased over the evaluation period. CAT is a H_2_O_2_-degrading enzyme that produces O_2_ and H_2_O molecules, and the higher activity of CAT suggested the ability of the cells to defend ROS by scavenging H_2_O_2_ molecules. However, the regulatory mechanism by CAT may not be efficient at fine-tuning the sensitive redox balances with low H_2_O_2_ concentrations [34]. SOD and POD were also responsible in detoxification process; the SOD enzyme prevents the cells from O_2_—induced radical damage through catalyzing the dismutation of the superoxide free radical to O_2_ and H_2_O_2_. Increased SOD and POD activities induced by the garlic root exudates could provide more appropriate protection against the oxidative damage caused by O^2−^ radicals [19]. Interestingly, the stimulation of SOD, POD, and PAL activities by exposure to garlic root exudates/garlic extract exhibited a synergistic effect, as confirmed by our earlier findings. Antioxidant capacities remain functional and membrane lipid peroxidation damage could be reduced if the pepper/garlic ratio does not exceed 1:2 or 1:4 [19]; similarly, above 300 µg mL^−1^ garlic bulb extract, the SOD activity drastically decreased and the malondialdehyde (MDA) content increased [20]. To explain these results, one possibility might be due to the incorporation of a greater number of garlic bulbs, which exerts a greater phytotoxic effect. Another assumption could be related to the genotypic difference and autotoxicity, as well as allelopathic potential.

The defense mechanism against ROS damage was further evaluated by the activation of non-enzymatic molecules, such as ascorbate and glutathione components. Ascorbate (ASA) and glutathione (GSH) are well-described stress indicators that have an interaction with a cysteinyl thiol group and are typically engaged in maintaining the complex regulation of several physiological processes. Through exposure to multifarious eco-stresses in plants, ASA and GSH responses have been reported regarding the detoxification of unfavorable ROS forms, stress tolerance, intracellular redox signaling, and organ development [42]. Our results are corroborated with those significant changes in thee redox status of the ASA and GSH pool, and the antioxidant adaptation plays a fundamental role in the regulation of ROS scavenging. In summary, our data proved that the accumulation of ROS induced by garlic root exudates activated the enzymatic and non-enzymatic reactions in cucumber leaves.

### 3.3. Effect on Photosynthetic Gas Exchange

The best-characterized phytotoxic mechanisms induced by allelochemicals are the inhibition of photosynthesis and oxygen evolution through interactions with components of photosystem II [10]. The inhibition of photosynthetic capacity, intracellular CO_2_ concentration, and stomatal conductance was frequently observed under allelopathic stress, and photo inhibition investigations indicated significantly greater physiological stress [43,44]. Meanwhile cucumber showed a significantly greater reduction in Gs than other analyzed variables when exposed to treatments (Figure 6B). An increase or decrease in Ci levels could be associated with stomatal and non-stomatal factors that are responsible for the reduction in Pn [45]. During stomatal limitation, a decrease in stomatal conductance coincides with a decline in the CO_2_ assimilation rate; in contrast, a reduced stomatal conductance and an increased intracellular CO_2_ concentration is characterized by a non-stomatal limitation. In our study, we found a decrease in stomatal conductance with a decrease in intracellular CO_2_ concentration, suggesting the decrease in photosynthetic rate induced by allelochemicals was at least partly due to stomatal closure [46]. Pn was largely dependent on the stomatal aperture, where stomatal base responses are considered to be the major contributor to the inhibition of photosynthesis [47]. Our results show that the proportion of the stomatal closures, together with the loss of leaf turgor, progressively leads to showing stress symptoms, followed by parallel decreases of the net photosynthesis rate in cucumber caused by allelopathic agents [46]. It is noteworthy that plant maturity also affected the photosynthesis alteration and the extent of phytotoxicity triggered by allelochemicals periodically shifted in a low–moderate–high fashion with plant aging. To the best of our knowledge, the direct inhibitory effect of allelopathic agents on photosynthesis has not yet been demonstrated, and the mechanistic involvement of phototoxins in photosynthesis the process has not been confirmed until now. In our study, the rates of Pn, Gs, Ci, and Tr significantly decreased after exposure to garlic root exudates, which might be the consequences of ROS generation. The disruption of pigment loss and leaf photosynthetic capacity is a key phytotoxic action of allelochemicals and ROS could be responsible for such physiological abnormalities [48].

### 3.4. Photosystem II (PSII) Inhibition: Limiting ROS Mediated Damage

Evaluation and serial monitoring of the photosynthetic performance during abiotic stress is an important component of photochemistry. Under such circumstances, chlorophyll fluorescence is a useful tool to assess the physiological status of plants, as well as abrupt changes in photosystem II (PSII) [49]. Photoinhibition of photosynthesis is typically characterized as a reduction in the quantum yield of photosystem (PS) II photochemistry and a decrease in chlorophyll fluorescence. In the present study, chlorophyll fluorescence data indicated there was a significantly greater physiological stress compared to the control in the beginning but intra-treatment variations of garlic-exudate-treated plants were not significantly different at later stage of exposures (Table 2). The photoinhibition occurred by measuring the maximal photochemical efficiency of open PSII reaction centers in the dark-adapted state (Fv/Fm) and quantum efficiency of the open PSII reaction centers in the light state (Fv’/Fm’). Both ratios significantly reduced as they were affected by garlic exudates. An observed reduction could have resulted in the impairment of thylakoid membranes, especially those of PSII, and the inhibition of energy transfers from antenna molecules to reaction centers can lead to lower Fv/Fm and Fv’/Fm’ ratios [50]. Sorgoleone is a well-documented inhibitor of PSII, where its potent response has been reflected in the down-regulation of the quantum yield of ΦPSII electron transport. The typical PSII inhibition by sorgoleone is one of the best-characterized mechanisms of allelochemicals, and other phytotoxins may have similar activity depending upon their mode of action at the cellular level [51]. Our results are in line with other findings suggesting that cucumber allelochemicals and their derivatives, such as cinnamic acid (CA), also act as photosystem inhibitors by reducing the photochemical efficiency of PSII (Fv/Fm) and the quantum yield of PSII, and caused damaged to the photosynthetic apparatus [7]. Garlic root exudates also decreased the level of quantum yield (ΦPSII) of photosystem II in cucumber. Reduction in the photochemical quantum production and its yield might be due to a decrease in the efficiency of the excitation energy trapping of the PSII reaction centers [52]. A similar declining response is observed in the qP level after treatment, and a reduction in the qP level might indicate that the balance between the excitation rate and electron transfer rate has changed, leading to a higher proportion of closed PSII reaction centers induced by allelochemicals [53,54]. Non-photochemical quenching (NPQ) represents the photoprotective mechanism of PSII by dissipating excess energy when plants are exposed to stress [55]. The NPQ value increased more than in the control due to garlic root exudates and this observation is similar to other findings regarding plants affected by allelochemical stress [56,57]. In contrast, the decay in NPQ value was also investigated in CA-treated lettuce plants. The drop in NPQ might be the result of a possible uncoupling of the thylakoid membranes induced by the high ion levels [7].

### 3.5. Metabolism of Carbohydrates

The partitioning of the metabolism of carbohydrates is strongly influenced by stress-related stimuli or other environmental cues, which can have deep developmental effects. The complex regulatory mechanism in the plant life cycle determines the allocation of carbohydrates between the different plant organs, reflecting the source–sink transition [58]. In this study, changes in the carbohydrates level significantly affected the roots and leaves tissues due to garlic root exudates. Leaf tissues progressively retained high amounts of soluble sugar, sucrose, and hexose contents, and the treatment affect was the opposite in root tissues. Sugar depletion in root tissues may reflect the poor sink activity due to a lower substrate availability in the storage of imported assimilates, and thus, various stress constraints disrupt the phloem integrity by impairing the transport function of the phloem, resulting in the source-to-sink supply of soluble sugars being decreased [59]. Stress presumably accelerates the imbalance of sugar allocation or its translocation in different compartments. Species-dependent variability ostensibly reveals the differences in the distribution and the activity of transport proteins and the enzymes that metabolize sugars [60]. Garlic root exudates mediating the increase in soluble sugar level could also explain the sugar function as an osmoprotectant and stabilize cellular membranes via a differential treatment response. Our results are in harmony with similar findings signifying that growth treated with allelochemicals could impose metabolic stress through the metabolism of soluble carbohydrates, which may challenge the integrity of the stress-induced source–sink relationship via the indirect interference of intra-specific allelopathic interaction [61,62,63].

Starch is a significant form of a carbohydrate reservoir. We observed starch depletion in both the root and leaf samples for the final stages, where the decrease in the amount of starch in the leaves was relatively higher than in the root organs (Figure 7D). Starch degradation could be related to an increase in the sugar concentration in leaves [64]. Several authors investigated the stress-induced metabolic changes in starch contents and suggested that these diurnal fluctuations could support the incidence of starch degradation into sugar accumulation [65].

Sucrose is also an important photo-assimilate that triggers essential metabolic events in response to diverse environmental signals [66], where a higher sucrose accumulation in leaves has been proposed as a signal molecule for plant immunity [67]. In our study, higher leaf sucrose levels in cucumber plants could be associated with increases in sucrose phosphate synthase (SPS) and sucrose synthetase–synthetic (SS-s), which show that allelopathic garlic exudates could modulate SPS and SS-s activities to induce sucrose accumulation from the source side (Figure 8B,D). SPS is the key regulatory enzyme that controls the carbon partitioning in the regulation of sucrose synthesis in stressed plants [68]. Another biosynthetic enzyme, SS-s, is widely observed in many plant species and the higher activity of that enzyme is involved in directing the carbon flow to cell wall synthesis [69]. The garlic-mediated decrease in sucrose formation in roots might be related to a reduction of the sink demand due to root growth limitation [58]. Similar to this, adverse stress toxicity impairs the sink activity earlier than source activity and causes modification of the source supply and sink demand by directly exerting an inhibitory effect on phloem sucrose loading and translocation, leading to a deficit in sucrose partitioning to the roots [70]. Correspondingly, it is also possible that a decrease of sucrose in cucumber roots mediated by the incorporation of garlic root exudates could be related to the inhibition of SPS activities.

In our study, the activities of both sucrose-cleaving enzymes showed a downward regulation pattern that may indicate higher sucrose biosynthesis at the source site rather than its greater degradation in sink tissues (Figure 8A,C). In general, we speculate that the garlic-induced sugar accumulation in the leaves might be associated with an increase in the net activity of sucrose-synthetic enzymes (SPS, SS-s). Sucrose synthetase–cleavage (SS-c) decreased markedly, which suggests that salinity-mediated decreases in both might reflect the reducing capacity of assimilates in the source-to-sink translocation, ultimately flagging the sink viability [71]. Phenolic acid also caused the modification of the carbohydrate status. Physiological actions of simultaneous exogenously applied phenolic constituents, such as putative ferulic acid (FA) and cinnamic acid (CA), has been reported regarding sucrose metabolism. The substantial sucrose accumulation was strongly associated with the reduction of hydrolytic enzyme activities [72].

However, the regulatory effects of allelopathic agents on the metabolism of carbohydrates in oxidative-stress-sensitive plant organs are still poorly understood and is still a matter of debate. However, one possibility suggests that the reduction of CO_2_ assimilation can involve the accumulation of carbohydrate metabolites and alter the source-to-sink translocation pattern in plants receiving an allelochemical treatment [45]. To verify this assumption, more definitive and systematic work is still required to prove that the “end product limitation is due to limited CO_2_ availability.”

## 4. Materials and Methods

### 4.1. Materials and Growth Conditions

The garlic cv. G064 was obtained from the horticulture department of Northwest A&F University, Yangling, Shaanxi, China. Fresh and uniform-sized garlic bulbs (almost 5 cm in diameter, 4 cm in height, and 45 g in weight) were selected and cultivated in sterilized perlite in plastic pots (40 cm × 30 cm × 12 cm) and incubated at 20 °C to yield a rooting culture. After completion of the specified growing duration to make sure maximum root development occurred, the garlic bulbs were uprooted at the fifth leaf stage at which its plantlets attained an approximately 5-cm root length and 50 cm in height. Harvested bulbs were rinsed with tap water to remove soil, sterilized with 0.1% potassium permanganate for 10 min, thoroughly rinsed with distilled water, and grown hydroponically in Hoagland nutrient solution for 30 days with cucumber seedlings.

The cucumber (*Cucumis sativus* L. cv. BoNai 13B) seeds were sterilized in hot water at 55 °C, stirred for twenty minutes, and finally rinsed three times with distilled water. The plant-growing media was commercially procured and contained peat moss for optimum growth and was autoclaved (125 Pa/121 °C) for 30 minutes and then poured into plastic pots (length × width × depth: 8 cm × 8 cm × 10 cm). The seeds were sowed in these pots, followed by shower irrigation, and then transferred to the growth chamber with a 25 °C/20 °C day/night temperature, 70% humidity level, and 12 h light period. Every two to three days, a shower irrigation was performed to ensure the uniform growth of seedlings. When the fourth leaves were fully expanded, the uniform-sized seedlings were transplanted to the hydroponic culture medium. These selected cucumber seedlings were rinsed with tap water to remove perlite, sterlized with 0.05% potassium permanganate for 20 min, and finally rinsed with distilled water and grown hydroponically in Hoagland nutrient solution for 30 days with garlic plants.

### 4.2. Experimental Design 

The experiment was conducted in April 2015 at the Horticultural Experiment Station (34°16′ N, 108°4′ W), College of Horticulture, Northwest A&F University, Yangling, Shanxi province, China. Each hydroponic system was mainly composed of a hard culture plastic pot of 45 cm × 35 cm × 15 cm (length × width × depth) and an air pump (220 V, 80 W) with a time switch. The air was periodically recycled by the pump (20 min/h). A total five treatments were proposed in the experiment in which two cucumber plants were co-cultured with 0, 5, 10, 15, and 20 garlic plants per pot, respectively. Therefore, the cucumber/garlic ratios (here after presented as C/G ratio) were 2:0 (control), 2:5, 2:10, 2:15, and 2:20 (as shown in Appendix A). Each treatment was replicated twelve times. Hoagland nutrient solution was formulated, and each pot contained 12 L of nutrient solution, where the culture solution was renewed every 7 days. The pH was adjusted to 6.2 ± 0.1 using 1 M H_2_SO_4_ or 1M NaOH. After 20, 28, and 36 days of treatment, the recent fully expanded leaves and roots were harvested and immediately transferred to the laboratory for the determination of related indicators.

### 4.3. Measurements

#### 4.3.1. Plant Growth

Leaf and root samples were simultaneously harvested on days 20, 28, and 36 after treatment. The plant dry weight was measured after oven drying at 80 °C for 72 h. The leaf area was measured using a hand-held leaf area meter (AM-350, ADC Bioscientific Ltd, Herts, England). The SPAD value of the cucumber leaves were measured using a SPAD-502 (Konica Minolta, Tokyo, Japan)

#### 4.3.2. Oxidative Damage Parameters

The oxidative stress-related parameters MDA and proline content in the leaves were determined according the procedures followed by Bu et al. [73]. The protective enzyme superoxide dismutase (SOD, EC 1.15.1.1), peroxidase (POD, EC 1.11.1.7), and catalase (CAT, EC 1.11.1.6) was determined according to Ding et al. [20]. The non-enzymatic antioxidant compounds of ascorbic acid (ASA) and glutathione (GSH) were determined using the method of Kampfenkel et al. [74].

#### 4.3.3. Gas Exchange Measurements 

Gas exchange parameters were measured using a portable photosynthesis system (LI-6400, LI-COR Inc, Lincoln, Nebraska, USA). The photosynthetic rate was measured at 800 μmol/mol CO_2_, 25 °C, 70% relative humidity, and light intensity of 1000 μmol m^−2^ s^−1^. The stomatal limitation (Ls) was calculated using Ls = 12Ci/Ca (Ci: intercellular CO2 concentration; Ca: ambient CO_2_ concentration).

#### 4.3.4. Chlorophyll Fluorescence Parameters 

Chlorophyll fluorescence was measured using the second fully expanded leaf from the apex at 25 °C using a chlorophyll fluorometer (PAM-2500, Walz, Effeltrich, Germany). The minimal fluorescence of leaves (F_0_) was measured under a weak pulse of modulating light (<0.1 μmol m^−2^ s^−1^, 600 kHz) after a dark adaptation for 30 min, and the maximal fluorescence (Fm) was induced using a saturating flash (8000 μmol m^−2^ s^−1^, 20 kHz) applied over 0.8 s. The maximal quantum efficiency of PSII was determined using Fv/Fm, where Fv is the difference between F_0_ and Fm. An actinic light source (600 μmol m^−2^ s^−1^) was then applied to achieve a steady-state photosynthesis and to obtain the Fs (steady-state fluorescence yield), after which, a second saturation pulse was applied for 0.7 s to obtain F’m (light-adapted maximum fluorescence). Fluorescence parameters were measured using a PAM-2500 based on the dark-adapted and light-adapted fluorescence measurements. The quantum efficiency of PSII (ΦPSII) and the efficiency of the excitation capture by open PSII centers were calculated using (F’m ± Fs)/F’m and F’v/F’m, respectively [75]. Photochemical quenching (qP) was calculated using (F’m ± Fs)/(F’m ± F_0_) [76].

#### 4.3.5. Carbohydrate Content Measurements

Freeze-dried samples were used for the determination of the carbohydrate content. Sucrose, starch, and hexose (glucose and fructose) content were determined using a modified phenol-sulphuric acid method. One hundred milligrams of sample was extracted overnight in 25 mL 80% ethanol (*v/v*) and the supernatant was analyzed for hexose, sucrose, and total sugars. The residue was boiled for 3 h in 10 mL 2% HCl (*v/v*) and the supernatant was analyzed for starch content.

#### 4.3.6. Enzyme Activity of Carbohydrate Metabolism Measurements

Sucrose phosphate synthase (SPS 2.4.1.14) and sucrose synthase (SS 2.4.1.13) were extracted at 0 ± 4 °C in accordance with Lowell et al. [77]. SPS activity was assayed at 37 °C using the method of Zhu et al. [78]. The reaction mixtures (70 mL) contained 50 mM HEPES-NaOH (pH 7.5, HEPES, 2-[4-(2-hydroxyethyl)piperazin-1-yl] ethanesulfonic acid), 15 mM MgCl2, 1 mM EDTA (Ethylenediaminetetraacetic acid), 5 mM NaF, 6 mM UDPG (uridine diphosphate glucose), 4mM Fru 6-P, 20 mM Glc 6-P, and 20 mL crude enzyme. Reaction mixtures were incubated for 30 min at 37 °C and the incubation was terminated with 70 mL of 5 M NaOH. Tubes were placed in boiling water for 10 min to destroy any unreacted fructose or fructose 6-P. After cooling, 1 mL of a mixture of 0.14% anthrone in 13.8 M H_2_SO_4_ was added and incubated at 40 °C in a water bath for 20 min. The color development of the cooled solutions was measured at 620 nm and the SPS activity was calculated. SS was assayed in both the synthetic (SS-s) and cleavage (SS-c) directions using the method of Lowellet [77]. Reaction mixtures (70 mL) for SS synthetic directions contained 80 mM HEPES (pH 8.5), 5 mM KCN, 5 mM NaF, 100mM fructose, 15 mM UDPG, and 20 mL desalted extract. Other assays used identical conditions to those for SPS. The mixtures (490 mL) used for sucrose cleavage contained 80 mM MES (pH 5.5, 2-(N-Morpholino)ethanesulfonic acid), 5 mM NaF, 100 mM sucrose, and 5 mM UDP. Reactions proceeded for 30 min at 30 °C and were terminated by the addition of 490 mL DNS reagent (3,5-dinitrosalicylic acid method). Tubes were heated in boiling water for 5 min. After cooling, color development was measured at 520 nm. Acid invertase (AI) was extracted as described by Zhu et al. [78]. Activity was assayed in a reaction mixture that consisted of 4% sucrose, 50mM sodium acetate buffer (pH 4.5), and an aliquot of enzyme solution in a total volume of 1 mL. The reaction mixture was incubated at 30 °C for 15 min. The reducing groups formed in the reaction mixture were measured in accordance with Entinana et al. [79].

### 4.4. Statistical Analyses 

The experiment was accomplished by using a completely randomized design (CRD). Values for all data were represented as the mean ± SE of four independent replicates. Analysis of variance (ANOVA) was performed to determine significant differences according to Fisher’s least significant difference (LSD) multiple range test at *p* < 0.05 using STATISTICA 8.0 software (StatSoft Inc. Tulsa, OK, USA). 

## 5. Conclusions

The incorporation of garlic root exudates in a small fraction contributed as growth regulators, as previously defined in our study, provided new allelopathic insight into exploring their quantitative strength by focusing on possible cropping obstacles in cucumber production. Present findings revealed that garlic root exudates triggered the oxidative stress due to severe lipid peroxidation, as well as proline and ROS induction, in cucumber during certain developmental stages. The result of these effects appeared in the reduction of photosynthetic performance that led to a constitutive accumulation of carbohydrates in the leaves and a decreased assimilate export to the roots, resulting an in imbalance in the source-to-sink photo-assimilate flow, which may account for the feedback inhibition of photosynthesis. We suggested that a high proportion of garlic root exudates in the cropping system might perform as strong allelopathic agents, which directly or indirectly affect the cucumber morpho-physiological indices. These findings may have practical implications for understanding the allelopathic inhibition mechanisms and further guide the development of new sustainable measures in the mitigation of alleloapthic stress.

## Figures and Tables

**Figure 1 plants-09-00045-f001:**
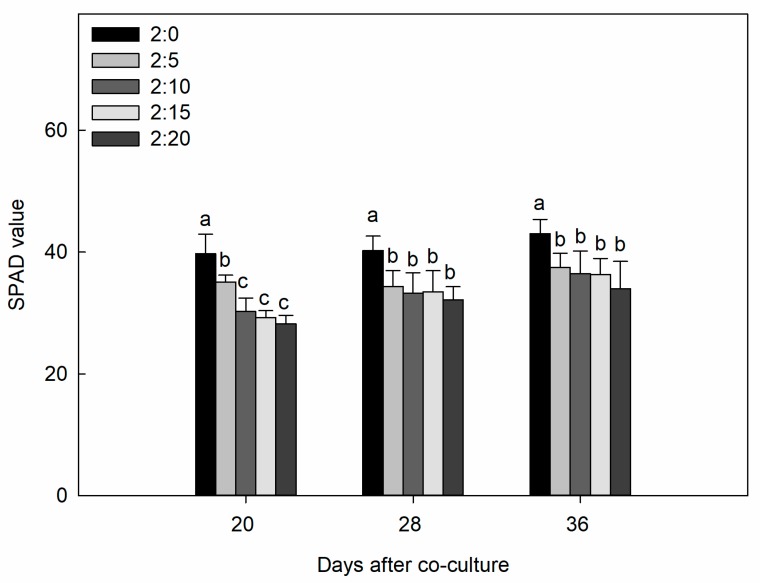
Effect of different C/G ratios on the SPAD (soil plant analysis development) value of cucumber plants in the hydroponic co-culture system. Data are the means ± SE of four replicates. The letters “a,” “b,” “c,” and “d” indicate that the values were statistically different at *p* < 0.05.

**Figure 2 plants-09-00045-f002:**
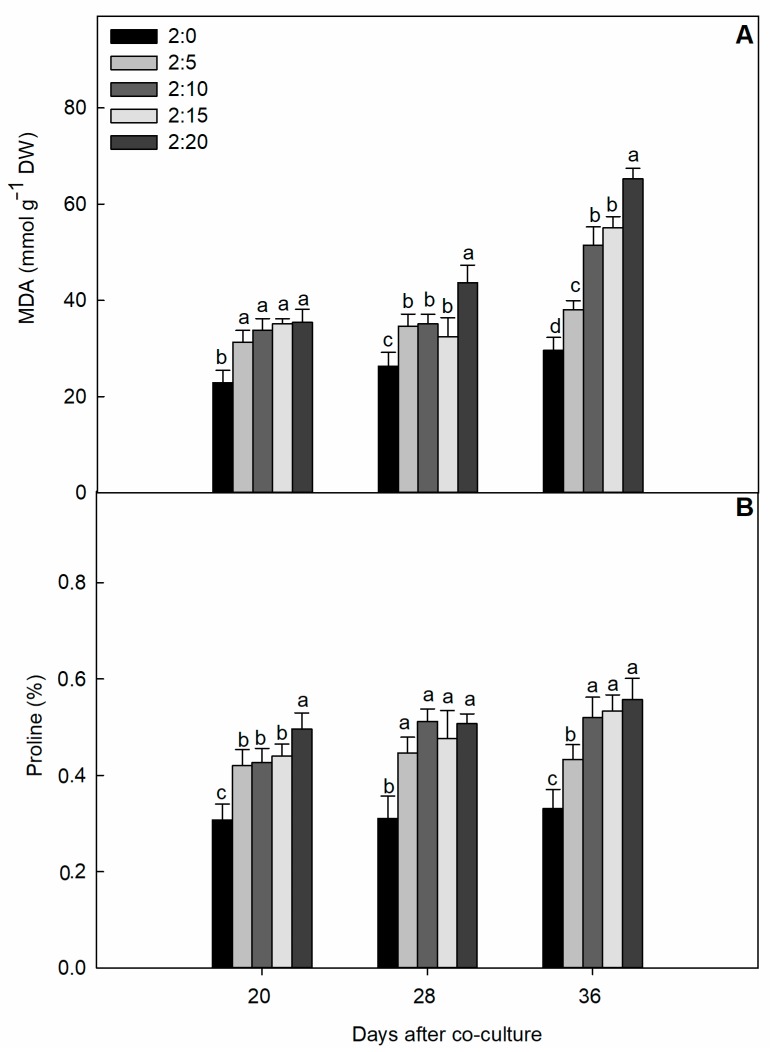
Effect of different C/G ratios on the content of malondialdehyde (MDA) (**A**) and proline (**B**) of cucumber leaves in the hydroponic system. Data are the means ± SE of four replicates. The letters “a,” “b,” “c,” and “d” indicate that the values were statistically different at *p* < 0.05. DW, dry weight.

**Figure 3 plants-09-00045-f003:**
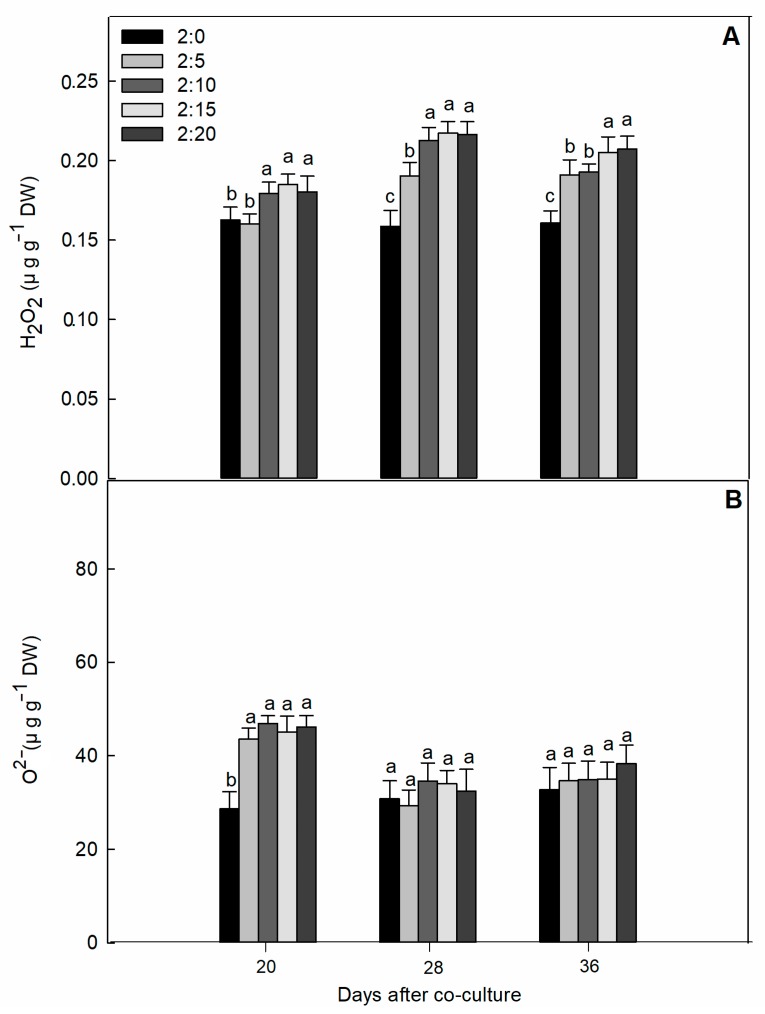
Effect of different C/G ratios on the content of H_2_O_2_ (**A**) and O^2−^ (**B**) of cucumber leaves in the hydroponic system. Data are the means ± SE of four replicates. The letters “a,” “b,” “c,” and “d” indicate that the values were statistically different at *p* < 0.05.

**Figure 4 plants-09-00045-f004:**
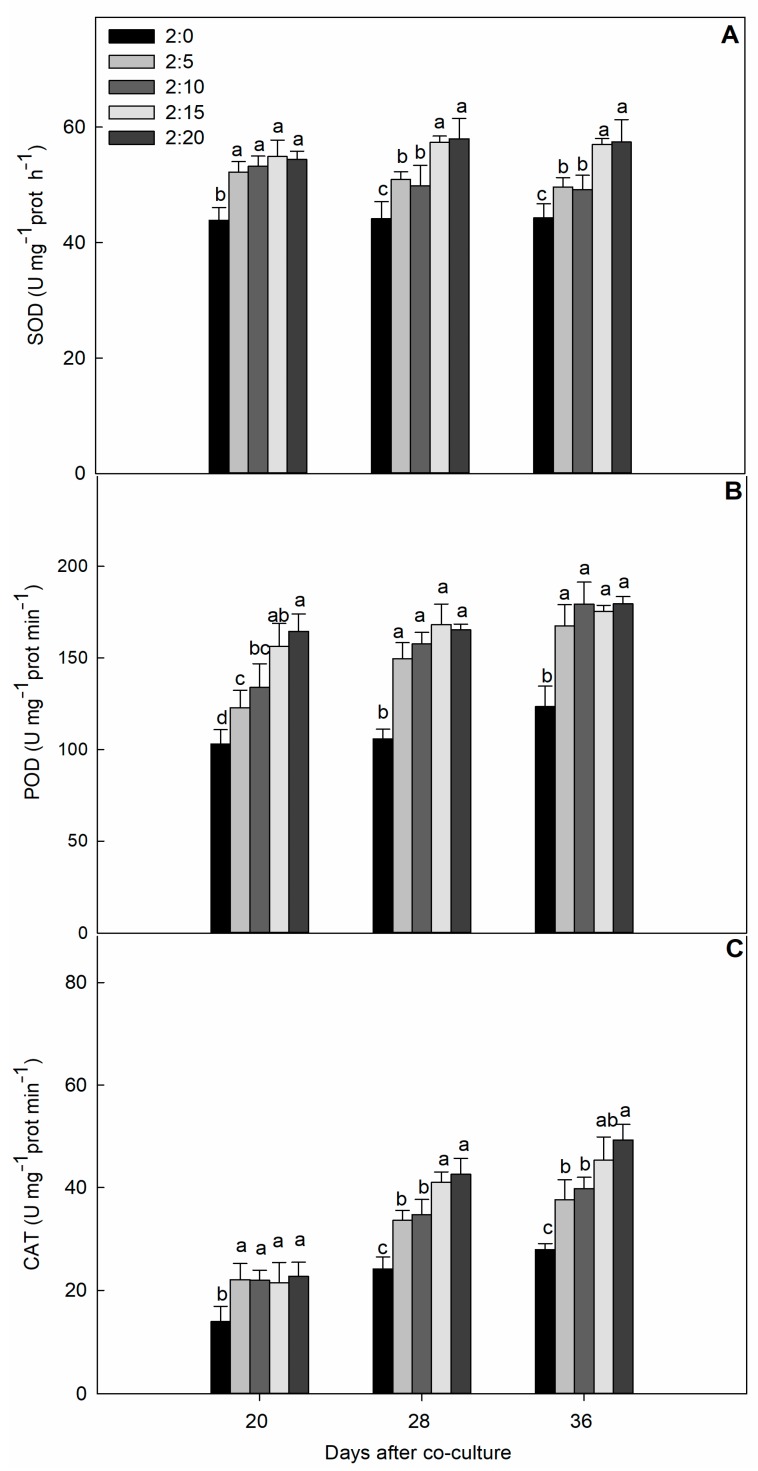
The effect of different C/G ratios on the SOD (**A**), POD (**B**), and CAT (**C**) activities of cucumber leaves in the hydroponic system. Data are the means ± SE of four replicates. The letters “a,” “b,” “c,” and “d” indicate that the values were statistically different at *p* < 0.05.

**Figure 5 plants-09-00045-f005:**
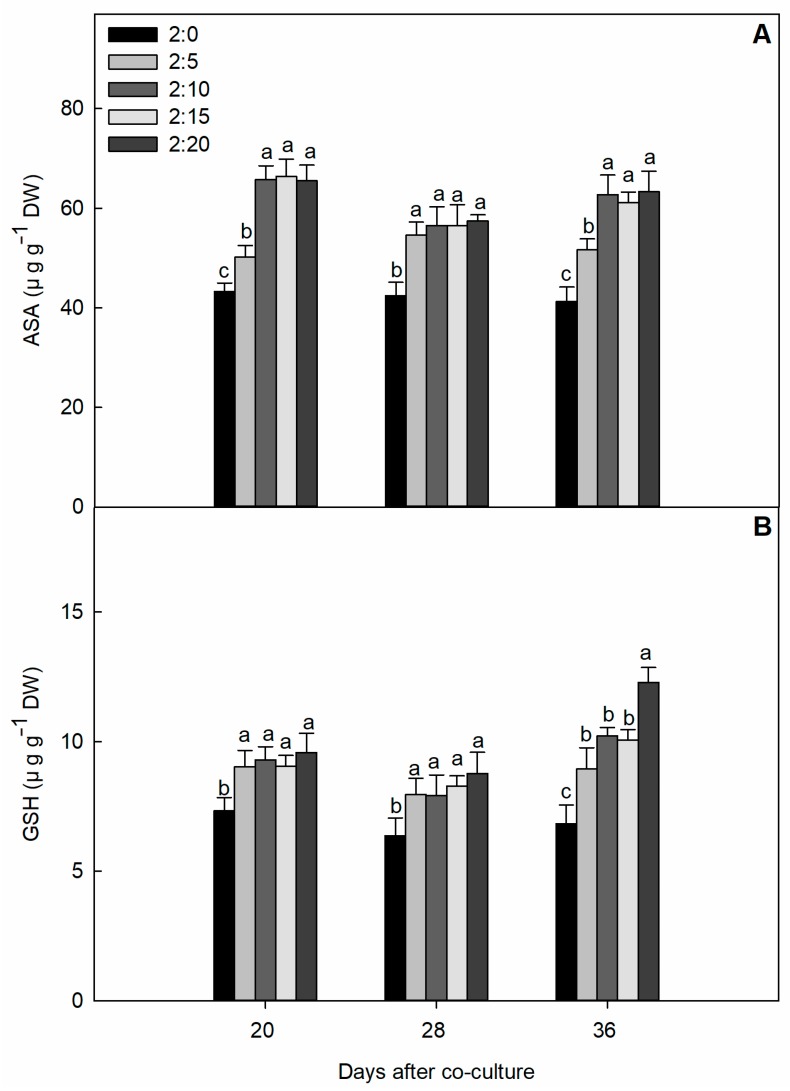
The effect of different C/G ratios on the ASA (ascorbic acid, (**A**)) and GSH (glutathione, (**B**)) content of cucumber leaves in the hydroponic system. Data are the means ± SE of four replicates. The letters “a,” “b,” “c,” and “d” indicate that the values were statistically different at *p* < 0.05.

**Figure 6 plants-09-00045-f006:**
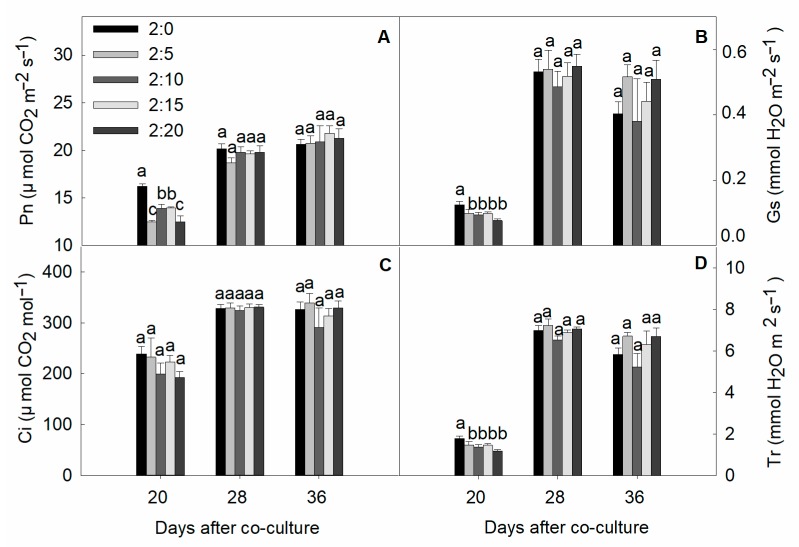
Effect of different C/G ratios on the gas-exchange parameters of cucumber plants in the hydroponic co-culture system. (**A**) Pn—net photosynthetic rate, (**B**) Gs—stomatal conductance, (**C**) Ci—intercellular CO_2_ concentration, and (**D**) Tr—transpiration rate. Data are the means ± SE of four replicates. The letters “a,” “b,” “c,” and “d” indicate that the values were statistically different at *p* < 0.05.

**Figure 7 plants-09-00045-f007:**
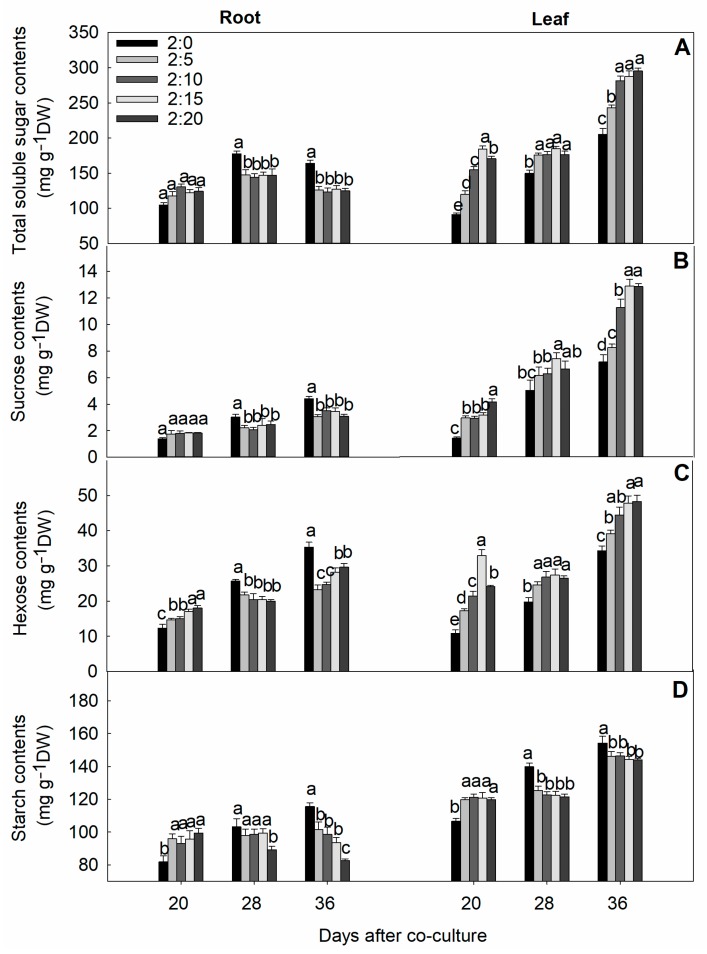
Effect of different C/G ratios on the contents of total soluble sugar (**A**), sucrose (**B**), hexose (**C**), and starch (**D**) in cucumber plants in the hydroponic co-culture system. Data are the means ± SE of four replicates. The letters “a,” “b,” “c,” and “d” indicate that the values were statistically different at *p* < 0.05.

**Figure 8 plants-09-00045-f008:**
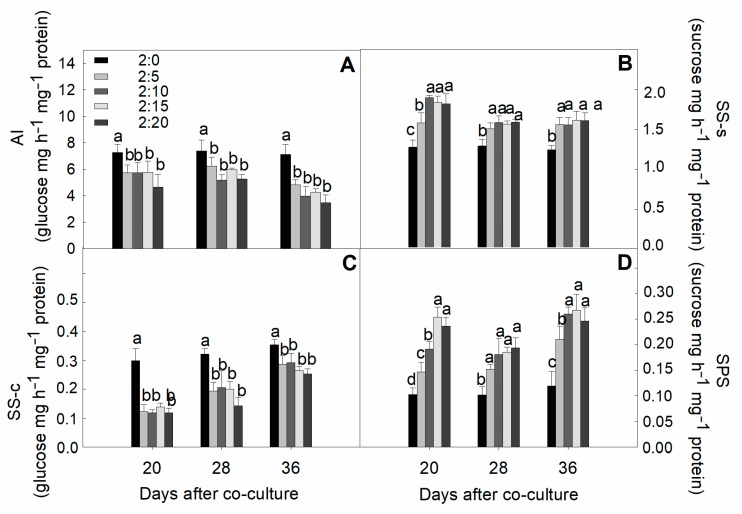
Effect of different C/G ratios on the activities of enzymes involved in sucrose synthesis and catabolism in cucumber plants in the hydroponic co-culture system. (**A**) acid invertase (AI), (**B**) sucrose synthetase-synthetic (SS-s), (**C**) sucrose synthetase-cleavage (SS-c), and (**D**) sucrose phosphate synthase (SPS). Data are the means ± SE of four replicates. The letters “a,” “b,” “c,” and “d” indicate that the values were statistically different at *p* < 0.05.

**Table 1 plants-09-00045-t001:** Effect of different cucumber/garlic ratios (C/G) on the growth of cucumber plants in the hydroponic co-culture system.

C/G Ratios	Increase of Shoot Dry Mass Per Day (g·plant^−1^·day^−1^)	Increase of Root DRY Mass Per Day (g·plant^−1^·day^−1^)	Increase of Total Dry Mass Per Day (g·plant^−1^·day^−1^)	Increase of Leaf Area Per Day (cm^2^·plant^−1^·day^−1^)
2:0	1.021 ± 0.052 a	0.135 ± 0.016 a	1.156 ± 0.026 a	12.911 ± 0.217 a
2:5	0.930 ± 0.027 b	0.125 ± 0.028 ab	1.055 ± 0.018 ab	10.875 ± 0.202 b
2:10	0.852 ± 0.023 bc	0.109 ± 0.033 b	0.961 ± 0.033 b	9.672 ± 0.352 c
2:15	0.825 ± 0.017 c	0.114 ± 0.026 b	0.939 ± 0.021 bc	10.014 ± 0.110 c
2:20	0.792 ± 0.036 c	0.117 ± 0.017 b	0.909 ± 0.037 c	9.688 ± 0.177 c

Data are the means ± SE of four replicates. The letters “a,” “b,” “c,” and “d” indicate that the values were statistically different at *p* < 0.05 (ANOVA and Fisher’s LSD (Fisher’s least significant difference) multiple range test).

**Table 2 plants-09-00045-t002:** Effect of different C/G ratios on the chlorophyll fluorescence of cucumber plants in the hydroponic co-culture system.

Days after Co-Culture	C/G Ratio	Fv/Fm	Fv’/Fm’	ΦPSII	qP	NPQ
20	2:0	0.759 ± 0.013 a	0.692 ± 0.005 a	0.614 ± 0.011 a	0.888 ± 0.017 a	0.230 ± 0.012 b
	2:5	0.705 ± 0.005 b	0.614 ± 0.007 b	0.572 ± 0.016 b	0.749 ± 0.019 b	0.323 ± 0.027 a
	2:10	0.711 ± 0.008 b	0.632 ± 0.044 b	0.531 ± 0.056 b	0.727 ± 0.044 b	0.335 ± 0.030 a
	2:15	0.714 ± 0.014 b	0.625 ± 0.006 b	0.578 ± 0.020 b	0.756 ± 0.024 b	0.355 ± 0.022 a
	2:20	0.700 ± 0.016 b	0.622 ± 0.023 b	0.509 ± 0.025 c	0.781 ± 0.066 b	0.372 ± 0.027 a
28	2:0	0.746 ± 0.012 a	0.671 ± 0.037 a	0.573 ± 0.023 a	0.853 ± 0.030 a	0.244 ± 0.025 a
	2:5	0.736 ± 0.019 a	0.645 ± 0.023 a	0.596 ± 0.053 a	0.806 ± 0.058 a	0.224 ± 0.049 a
	2:10	0.765 ± 0.025 a	0.692 ± 0.031 a	0.621 ± 0.038 a	0.897 ± 0.020 a	0.249 ± 0.038 a
	2:15	0.760 ± 0.014 a	0.681 ± 0.054 a	0.603 ± 0.054 a	0.884 ± 0.018 a	0.265 ± 0.029 a
	2:20	0.753 ± 0.038 a	0.659 ± 0.049 a	0.580 ± 0.047 a	0.880 ± 0.017 a	0.263 ± 0.063 a
36	2:0	0.772 ± 0.008 a	0.673 ± 0.030 a	0.591 ± 0.045 a	0.875 ± 0.033 a	0.386 ± 0.035 a
	2:5	0.741 ± 0.031 a	0.654 ± 0.027 a	0.554 ± 0.026 a	0.823 ± 0.052 a	0.403 ± 0.057 a
	2:10	0.782 ± 0.007 a	0.678 ± 0.021 a	0.577 ± 0.047 a	0.848 ± 0.043 a	0.425 ± 0.047 a
	2:15	0.772 ± 0.003 a	0.672 ± 0.013 a	0.537 ± 0.030 a	0.906 ± 0.058 a	0.396 ± 0.091 a
	2:20	0.770 ± 0.005 a	0.672 ± 0.024 a	0.555 ± 0.032 a	0.884 ± 0.019 a	0.399 ± 0.123 a

Data are the means ± SE of four replicates. The letters “a,” “b,” “c,” and “d” indicate that the values were statistically different at *p* < 0.05 (ANOVA and Fisher’s LSD multiple range test). Fv/Fm (photochemical efficiency of PSII), Fv’/Fm’ (capture efficiency of PSII), ΦPSII (quantum yield of PSII photochemistry), qP (photochemical quenching coefficient), NPQ (non-photochemical quenching.

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
