# Peer review of "An Allelopathic Role for Garlic Root Exudates in the Regulation of Carbohydrate Metabolism in Cucumber in a Hydroponic Co-Culture System"

_plants, 2019, doi:10.3390/plants9010045_

Round 1

Reviewer 1 Report

Dear Authors

Although the garlic is considered to have a strong positive effect on the growth and yield of receptors under soil cultivation conditions, the mechanism is still unclear. In this study, authors used the hydroponic co-culture system to investigate the direction action of garlic to cucumber. Results indicated that garlic root exudates have negative effect on cucumber through the oxidative stress imposition and the imbalanced the source-to sink photo-assimilate flow. This manuscript is well organized, comprehensively described, the results appear sound and the discussion is well done. However, some data and experimental design are need to be added and discussed. I recommend this manuscript to be accepted after minor revision.

According to your design, garlic plants were co-culture with cucumber in the same container. Are there any pictures can be showed in the manuscript? Please added. When those plants co-culture in the same size container, the growth inhibition may be due to the nutrients competition rather than allelopathic effects. How to avoid this competition? In addition, when they grow together, are there any shade avoidance happened? Please discuss this issue.

Author Response

Thanks for your valuable comments, please see the attachment file.

Reviewer 2 Report

The article proposed by Haiyan Ding is well written and well organised. I have some minor suggestion to improve the quality of the paper:

explain the values reported in table 1. Are they mean and standard deviation? In addition, I suggest to check all the digit numbers, which should be approximated considering the variability of the analytical methods.  Why the authors analysed MDA? There are other markers of oxidative stress, such as carbonyl, isoprostanes and prostanoids. Please considere these artcile and explain the reasons of this choice. In all the figures, please explain the number of replicates.

Regards.

Author Response

(The authors gave the same response as above.)
